# Learning to play slot cars and Atari® 2600 games in just minutes

## Abstract

Machine learning algorithms for controlling devices will need to learn quickly, with few trials. Such a goal can be attained with concepts borrowed from continental philosophy and formalized using tools from the mathematical theory of categories. Illustrations of this approach are presented on a cyberphysical system: the slot car game, and also on Atari 2600 games.

## 1 Introduction

There is a growing need for algorithms that control cyberphysical systems to learn with very little data how to operate quickly in a partially-known environment. Many reinforcement-learning (RL) solutions using neural networks (NN) have proved to work well with emulators, for instance with the Atari[1] 2600 games (Mnih et al., 2015), or with real systems such as robots (Lange et al., 2012). However, these state-of-the-art approaches need a lot of training data, which may not be obtainable within the allowed time frame or budget. This work thus started as an alternative approach to teach computers to learn quickly to perform as efficiently as the existing solution with approximately one percent of the training data, time, and computing resources.

We first review reinforcement learning methods for Markov Decision Processes (MDP) and Partially Observable MDP (POMDP). We then explain the motivation behind our continental-philosophy-inspired approach. We describe the two classes of problems on which we focus: the bijective case, which may lead to playing by imitating, and the category-based approach, which should lead to a more innovative behavior of the control algorithms. Both approaches rely on knowledge accumulated during previous experiences, as in Lifelong Machine Learning (Chen & Liu, 2016).

These two approaches are illustrated by results from both a commercial slot car game controlled by an 8-bit Arduino system, and from Atari 2600 video games running within the Arcade Learning Environment (ALE, see Bellemare et al. (2013)).

## 2 Related work

### 2.1 Slot cars

The development of Artificial Intelligence (AI) owes much to games, which have become one of the classical test-beds for algorithms. Slot car games, for instance, are used to evaluate the performance of decision-making systems. The image-processing, RL-based approach presented in Lange et al. (2012) estimates a car's position on the track thanks to a multilayer perceptron with convolutional layers. It controls the car by applying one out of four possible voltage levels. Training the perceptron takes twelve hours, and learning the control strategy needs another half an hour. The faster solution by Pusman & Kosturik (2013) to autonomous slot cars relies on added acceleration sensors and an embedded microcontroller to first create a map of curved and straight tracks. The control algorithm then sets the target velocity and controls it using a Phase-Locked Loop.

---

[1] Arduino™, Atari™, Breakout™, Frogger™, Mini™, Pac-Man™, Pong™, Scalextric™, Texas Instruments™ are trademarks of their respective owners and will be written without the trademark symbol for clarity in the remainder of this document.

Our system learns to rank with human players in less than a minute, without embedded sensors, for both known and unknown circuits, which correspond to the aforementioned bijective and category-based approaches. The sensors are a lap counter, and voltage and current from the track.

## 2.2 Video games with a focus on the Atari 2600 console

Video games also become increasingly useful for providing a cyber-physical representation of our environment. Indeed, realism turns out to be one of the main focuses for game and character designers (depending on the game intentions). Nevertheless, due to technical limitations and the need for easily described problems, older video-gaming systems, such as the Atari 2600 console, are the go-to systems. They provide a wide variety of situations ranging from mazes (Pac-Man-style games) and action games (such as Frogger) to ball-and-paddle games (Breakout, Pong). Although these different problems require varied strategies to be tackled by a standard human player, they all involve decision-making and, therefore, have been modeled as MDPs (Mnih et al., 2015).

This framework allows the implementation of many different methods. Some works, such as Mnih et al. (2015) use a rescaled picture of the playing area as an input for a deep Q-network (DQN) in order to select the best action available to the agent. Another possibility is the use of classic searching and planning methods in order to guide the agent, such as the Iterated Width algorithm (Lipovetzky et al., 2015) or tree search algorithms such as Monte-Carlo Tree Search (MCTS) (Pepels et al., 2014) to compute the best possible action for the agent. A less common method is Shallow Reinforcement Learning (Liang et al., 2016): although this relies on a simpler linear representation, it obtains results similar to those of the non-linear approaches. Finally, Apprenticeship Learning (Bogdanovic et al., 2015) and Inverse RL (Lee et al., 2014) can also be used to train a more human-like agent which is close to the aforementioned methods in terms of efficiency.

We will show that our system learns how to play unknown games in a few thousand frames with a score on par with or better than humans.

## 3 Continental-philosophy-based theoretical approach

The success of NN is partially due to the very large amount of data that is used, even if the programmer does not know exactly what happens in the NN (like in *black box* systems). Two problems are thus the huge quantity of data needed, and the training time required. Moreover, the attribution of good coefficients in the learning phase is very difficult – or impossible – to be interpreted, making validation very difficult. NN are able to learn and generalize, but we do not know exactly how.

To improve or complete the NN approach, we propose an approach that tries to explain and use explicitly how an AI can learn, extract features, categorize and generalize. To do this, we place the theoretical elements necessary for such high level abilities directly in the method. These abilities may also emerge in NN after many elementary computations (additions, multiplications, comparisons) occur at each artificial neuron. If this is what effectively happens in each biological neuron, we postulate that intelligence also consists in higher level intellectual operations. In other words, we do not want to reduce intelligence to basic computation – even if it is biologically the case. As we do not want higher level abilities to emerge (or not) after long training times, we explicitly place these high level abilities (such as categorizing and generalizing) directly in our theoretical framework.

Thus we can follow some aspects of Dreyfus' critique of AI presented in Dreyfus (1992). This author claims that AI researchers should focus more on what human intelligence is in itself and not only refer to the computer model: considering the brain as a computer, and intelligence as the use of software. More precisely, in the case of RL with NN: considering intelligence as a collection of elementary computations that organize themselves after much training to reach a reward goal. We postulate that it could have happened in such a way over the course of human development, but human intelligence has much evolved. It can produce categorization and generalization not merely for a simple reward, but for the goal of understanding. This is what our AI tries to do. Dreyfus also often refers to authors such as Heidegger, Husserl or Foucault, whose work later became known as continental philosophy. This name was given by analytic philosophers who were often Anglo-Saxon in origin, the earliest being Russell, Frege and Wittgenstein. Analytic philosophy received much influence from mathematical logic that emerged at the end of the 19th century. It tries to clarify philosophical issues by logical analysis, postulating that only philosophical statements verifiable

through empirical observations are meaningful (principle of logical positivism). This principle, according to analytic philosophers, is not respected by continental philosophers.

Continental philosophy includes a range of French and German doctrines from the 19th and 20th centuries: German idealism, phenomenology, existentialism (influenced by Kierkegaard and Nietzsche), hermeneutics, structuralism, post-structuralism, psychoanalytic theory and object-oriented ontology. These philosophies are all contrary to the analytic movement. If we had to project AI in this debate (analytic versus continental), we could say that Dreyfus criticizes early AI for favoring the analytic tradition and for neglecting the continental one. Of course machines are computers, and computing is closer in nature to logic than phenomenology, metaphysics or psychoanalysis. Continental philosophy, however, can perhaps help understand and describe what human intelligence is, especially for high level abilities, like learning, categorizing, generalizing and understanding. It could possibly then improve the quality and efficiency of human-intelligence-based AI.

To summarize, we propose to design our AI using an approach based on certain elements of continental philosophy. This philosophy is described, for lack of a more precise and widely-accepted definition, in terms of its opposition to analytical philosophy. In the next sections, we will propose some connections between this philosophy and existing mathematical theories.

### 3.1 Entities in space and time

We express the logic of our AI at the level of entities, and not at a sample or at a pixel level. In a way, this is similar to working at the morpheme level in structural linguistics, as defined by de Saussure (1916), which is the smallest meaningful unit of a language. That implies to start with an analysis of the sampled signals (in one or two dimensions) to detect entities. These entities are like our everyday life objects: tracks (straights and curves), cars, balls, paddles, walls. They are geometrically organized in a space and can be described by cartesian coordinates. We have defined a distance between them that measures how far two entities $E$ and $F$ are one from one another[2]. The data is collected at each sample time so that we can construct a timeline and provide an elementary cinematic newtonian model of the situation. This comes from a very old idea of developmental psychology (see for example the works of Piaget (1954)) that the child starts his cognitive development by the skill of experiencing the world through movement and senses (Piaget called it the *sensorimotor* stage). But the perceptive world is not a wild set of disordered primitive sensations. They are organized in objects (we call them *entities*) that take place in a space and can move during time[3]. Thus we do not want to take into account all the samples (voltage and current for the slot car, pixels' colors for images) as the fundamental level of knowledge. We shall try to organize them as soon as possible as entities that occur in space and time (and not wait for them to emerge, or not, after a very long learning process). These entities, like the objects of cognitive psychologists, have some properties : relative consistency, continuity of movement, permanence of existence and characteristics (sizes, color, shape). These properties are part of our approach, in the sense that our AI can look for rectangular entities with a particular position, speed and size[4]. In more complex games, this rectangular form approach could be too simple, but it is adequate for the Atari 2600 games that we study.

### 3.2 The Me-in-the-world

One of the main critiques formulated by Dreyfus against the old AI philosophy is the *epistemological assumption* that claims that all activities can be formalized in terms of predictive rules or laws. In this context, the learning phase consists of determining these rules (that is, their parameters). Then, the system has to apply them by looking for *objects* or general *characteristics* of the whole organization of samples, that are like those used in the learning phase. But what about new objects, never seen

---

[2]Note $E \cap F = R$ their intersection. If $R$ is empty, let $d(E, F) = Min\{d(M, M'), M \in E, M' \in F\}$. If $R$ is a rectangle with measure of the diagonal $d$, define $d(E, F) = -d$.

[3]The German philosopher Kant provided in Kant (1787) a theory of knowledge in which the subject sets up a framework, a form (in particular the space and the time) in which all sensations become organized.

[4]The object permanence (the fact that an object continues to exist even if it is not perceptible anymore) was studied by philosophers in early philosophy, then in the context of cognitive psychology, and now in AI frameworks like the one described by Chen & Weng (2004).

before, that could appear? Such a strict and trivial application of the epistemological assumption would lead to ignore them. It could also be a principle of precaution to ignore new objects.

On the other hand, there could be a principle of curiosity or adventure. Clever machines could be more efficient were they curious as explained in Pathak et al. (2017). Referring to the work of Alison Gopnik and Laura Schulz, developmental psychologists at Berkeley and at the Massachusetts Institute of Technology, respectively, it explains that babies naturally gravitate to objects that surprise them rather than to those they are used to, to achieve some extrinsic goal. An AI that only focuses on application of predictive rules will miss the advantages of curiosity. We will use this curiosity to further develop our AI in our next work.

If the epistemological assumption of usual AI could be useful for chemistry or physics, because they are *context-free*, it could be a contradiction in terms with psychology, and behavior understanding. Dreyfus argued that human problem solving depends rather on our background sense of the context, that is the natural feeling, understanding or intuition of what is important and interesting given a situation. The world is not just made of objects: it contains subjects. In particular, in the games we consider, there is a representative of what we call the "Me": the entity that is controlled by actions. This point of view allows for a more efficient approach than computing all the possible combinations of the available symbols. This is exactly what we do when we ask our AI to look as soon as possible for some important features (entities and the "Me"). Dreyfus (1990) referred to the Heideggerian concept of *Dasein* (which means "being there", for a human being confronted with such issues as personhood and mortality), which is a specific way of *Being-in-the-world* (another Heideggerian concept that considers it as a unity, saying that it is not appropriate to distinguish strictly between the Being and the world that it is in) (Heidegger, 1927).

In other words, one of the first things that our AI must do is to identify the "Me" from amongst all the listed entities. It is not an implicit potential result of a huge number of trainings, like in some RL processes. Moreover, being the "Me" does not mean only to be lead by actions. It also implies to struggle for life. We can say that the "Me" is driven by some *life impulses*, and that it is attracted to the good objects (that we call *friends*) and wants to go away from the bad ones (the *enemies*). Thus postulate that among all the entities, some are friends (those whose contact implies a reward or avoids loss of lives) and some are enemies (those whose contact implies loss of lives). The AI has to distinguish as soon as possible the friends versus the enemies of the "Me", without waiting for this to emerge from millions of trials. After that, the survival strategy is simple: try to meet the friends unless there is an enemy close to the "Me", in which case the first thing to do is to flee.

## 3.3 NON BIJECTIVE ANALOGIES

One of the most efficient tools that humans use to understand new situations is the ability to make *analogies* between past and present. For example, if the AI knows how to play the game Breakout, we expect that it will be able to transpose this ability to a (partially) analogous game, Pong. In particular, we hope to soon use mathematical tools to transpose a *policy* from one problem (for instance a game) to another. Such a theory is proposed by Bonet & Geffner (2015) for PONDP (Partially Observable Non Deterministic Problems).

The problem is that ideal situations where two problems have exactly the same number of states and isomorphic structures are very rare. Nevertheless, there are mathematical tools that can be used to identify non isomorphic structures like equivalence of categories in category theory (Mac Lane, 1998)[5]. The theory of category is a powerful tool in modern mathematics that appeared in the mid-20th century in topological and geometrical contexts, after the mathematical logic, based on set theory. If mathematical logic was a great source of inspiration for analytic philosophy, category theory could inspire and support continental ideas. The association between category theory and continental philosophy is proposed by Zalamea (2012) and we will follow this path in our work.

In a very simplistic way, we could say that if analytic philosophy *analyses* situations, by distinguishing states (or objects), continental philosophy provides *syntheses*, setting higher new levels of being (Beings, concepts, types). Whereas in set-theory-based logic, identification is reduced to identity and bijective relations, category theory provides richer descriptions of objects by the introduction

---

[5]Another perspective would be the use of the concept of elementary equivalence of model theory (Chang & Keisler, 1990) which could allow the AI to perform logical reasoning.

of arrows between objects, allowing new kind of identifications. The reader familiar with category theory may find obvious the rest of this paragraph. However, as most Machine Learning tools rely on set theory and not category theory, we try to illustrate below the added value of this mathematical framework. The reader is nevertheless referred to Mac Lane (1998) for a thorough and in-depth explanation of category theory. A category $\mathcal{C}$ is a collection of objects with arrows between some of them, so that we can compose them. It is something like an oriented graph. In $\mathcal{C}$, an arrow $a : A \rightarrow B$ is called an isomorphism if it is invertible, that is if there is an arrow $b : B \rightarrow A$, such that $ba = Id_B$ and $ab = Id_A$. If it is the case we say that the object $A$ and $B$ are isomorphic. The relation of isomorphism defines an equivalence relation on the collection of objects of $\mathcal{C}$. We note the quotient $\mathcal{C}/\simeq$. If $F : \mathcal{C} \rightarrow \mathcal{C}'$ is an equivalence of categories, it induces a real bijection $F' : (\mathcal{C}/\simeq) \rightarrow (\mathcal{C}'/\simeq)$ between the classes of isomorphic objects even if $F$ is not bijective. We do not identify the objects (or the states) of two situations one-to-one, we identify the types (or classes) of these states.

This process can be very useful in the context of observable problems. Let's consider two nonempty sets (of states) $\mathcal{C}$ and $\mathcal{C}'$ not necessary of the same cardinals. Let's suppose that we have two functions of observation $f : \mathcal{C} \rightarrow O$ and $f' : \mathcal{C}' \rightarrow O'$. Let's assume that they are surjective (if not, we can restrict $O$ and $O'$). The sets of observations $O$ and $O'$ will define some types of states. For each $o \in O$, we say that all the states $x \in \mathcal{C}$ that are observed as $o$ ($f(x) = o$), have the type $T_o$. This defines a natural equivalence relation $R_f$ on the set $\mathcal{C}$ : $\forall x, y \in \mathcal{C}, x \; R_f \; y$ if and only if $f(x) = f(y)$. In terms of categories, we put an invertible arrow between two objects $x$ and $y$ of $\mathcal{C}$ iff $x \; R_f \; y$ (iff stands for if and only if). This makes $\mathcal{C}$ a category, where all arrows are invertible and such that $\mathcal{C}/\simeq$ is exactly the quotient $\mathcal{C}/R_f$ : the set of types of states of $\mathcal{C}$. It is well known that the surjection $f : \mathcal{C} \rightarrow O$ induces a bijection $\tilde{f} : (\mathcal{C}/\simeq) \rightarrow O$ between the set of type and the set of observations. This is obvious since the types as been defined by the observations. $\tilde{f}$ is actually the inverse of $T : O \rightarrow (\mathcal{C}/\simeq), o \mapsto T_o$. We do exactly the same with $\mathcal{C}'$ and $f'$.

Suppose now, and this is very important, that the sets of observations $O$ ad $O'$ have the same cardinality by the means of a bijection $G : O \rightarrow O'$. Thus, we can define a bijection $F' = \tilde{f}'^{-1} \circ G \circ \tilde{f} : (\mathcal{C}/\simeq) \rightarrow (\mathcal{C}'/\simeq)$. This bijection between the sets of types can be induced by an equivalence of categories $F : \mathcal{C} \rightarrow \mathcal{C}'$ defined as follows : for every $x \in \mathcal{C}$, let's call $o = f(x)$ and chose an arbitrary $x' \in f'^{-1}(G(o))$, and define $F(x) = x'$. If $\mathcal{C}$ and $\mathcal{C}'$ do not have the same cardinality, $F$ has no chance to be bijective, but $F'$ is. $F$ sends every state $x$ to a state $x'$ of the "same" type (up to $G$). This is the way that we identify (not necessarily bijectively) $\mathcal{C}$ and $\mathcal{C}'$. Thus, if we have a strategy to play in $\mathcal{C}$, we can transpose it in $\mathcal{C}'$ thanks to $F$.

The use of the theory of category results in the ability to formalize a wide variety of games and situations. An illustration of this would be the ease with which a human player can switch from the Atari 2600 game Breakout to the very similar Pong. This ease can be transposed into the formalism of categories. However, even a much more concrete system such as the slot car described in section 2.1 and experimented on in section 4.1 can be transcribed into the formalism of categories[6].

Let us define the following sets:

- $\{\mathcal{C}, \mathcal{C}'\}$ is the set of categories (one per configuration of the track).
- $\{N, N'\}$ is the number of sections per configuration of the track.
- $\{s, s \in [1, N]\}$, $\{s', s' \in [1, N']\}$ are the possible locations of the car on the circuit. The location is obtained by counting the number of sections the car has passed in its current lap. We note $(u, i)_s$ (Resp. $(u', i')_{s'}$) the voltage and current measured when the car crosses section $s$ (Resp. $s'$) Let $1 \leq s_0 \leq N$ (Resp. $1 \leq s_0' \leq N'$) be the current position of the car in configuration $\mathcal{C}$ (Resp. $\mathcal{C}'$).
- Let $k'$ be a straight section and $l'$ a curve section of $\mathcal{C}'$.
- The player influences $(u, i)_s$ with the controller, which leads to the policy $\pi$ defined by (1).

$$\pi(s) = \begin{cases} (u', i')_{k'}, & \text{if } s \text{ is a straight line} \\ (u', i')_{l'}, & \text{otherwise} \end{cases} \tag{1}$$

---

[6]We are aware that sets and equivalence relations could be enough to formalize these toy models. However, we do believe that category theory is the good theoretical framework to be used for further more complex developments with several types of observations, with formalizations of actions on states by arrows in a graph.

We want to *identify* $\mathcal{C}$ and $\mathcal{C}'$, to transpose the policy $\pi$ from $\mathcal{C}$ to $\mathcal{C}'$. The states of $\mathcal{C}$ are the locations $s$ of the car in the circuit. Similarly, the states of $\mathcal{C}'$ are the $s'$. If $N = N'$ and $s_0 = s'_0$, we can define a bijection between $\mathcal{C}$ and $\mathcal{C}'$ and easily transpose $\pi$. But if $N \neq N'$ or $s_0 \neq s'_0$ it is impossible to define such a bijection.

Nevertheless, if we turn $\mathcal{C}$ and $\mathcal{C}'$ into categories by defining some arrows, we will be able to define an equivalence of categories $F : \mathcal{C} \to \mathcal{C}'$. To define these arrows, let's use the observable $f$ defined on the states $s$ of $\mathcal{C}$ as follows $f : \mathcal{C} \to \{1, 2\}$ with $f = h \circ g$ where $g$ and $h$ are such that $g(s) = (u, i)_s$ and $h((u, i)_s) = 1$ if $s$ is a curve, and 2 otherwise. We define $f'$ on the $s'$ of $\mathcal{C}'$ the same way, i.e. $f' : \mathcal{C}' \to \{1, 2\}$ with $f' = h' \circ g'$ and $g'$ and $h'$ playing the same roles as $g$ and $h$ on the states of $\mathcal{C}'$.

We can put an invertible arrow between two states of $\mathcal{C}$ iff they have the same image by $f$, and an invertible arrow between two states of $\mathcal{C}'$ iff they have the same image by $f'$. We then define $F : \mathcal{C} \to \mathcal{C}'$ by equation (2).

$$F(s_0) = \begin{cases} l', & \text{if } f(s_0) = 1 \\ k', & \text{if } f(s_0) = 2 \end{cases} \tag{2}$$

It is easy to see (if the exact definitions are known) that (2) is an equivalence of categories that allows to transfer $\pi$ from $\mathcal{C}$ to $\mathcal{C}'$. $F$ induces a bijection $F'$ between the sets of classes (or types of position): $F' : (\mathcal{C}/\simeq) \to (\mathcal{C}'/\simeq)$ where $(\mathcal{C}/\simeq)$ is $\{\mathscr{C}, \mathscr{S}\}$ and $(\mathcal{C}'/\simeq)$ is $\{\mathscr{C}', \mathscr{S}'\}$. We finally obtain $F'(\mathscr{C}) = \mathscr{C}'$ (types of curves) and $F'(\mathscr{S}) = \mathscr{S}'$ (types of straights).

This example of systematic categorization and generalization proves that we do not work at the level of states but that type of states are considered instead.

## 4 EXPERIMENTAL SETUP AND RESULTS

Results of this approach are presented for a cyberphysical system: a slot car circuit, and for a simulated system: Atari 2600 video games.

### 4.1 SLOT CAR

The focus on the slot car experimental setup arose from the need to validate the approach on a cyberphysical system. With its imperfect actuators such as a brushed, direct-current (DC) motor, imperfect contacts such as metallic brushes on strips, it allowed us to evaluate the approach while dealing with a wide range of signals from a real system. Moreover, its wide availability and low cost allowed to duplicate the test-bed so as to widen the span of the validations. On the other hand, the configuration is simple, as there is only one entity with dynamic behavior: the "Me' is the slot car. The enemies are located at unknown curvilinear abscissas where a high velocity is detrimental to the "Me".

#### 4.1.1 SETUP AND IMPLEMENTATION OF THE THEORY

The setup is based on a Scalextric MINI Challenge Set C1320T. We have replaced the mechanical lap counter by a digital omnipolar Hall effect sensor DVR5033 from Texas Instruments. The current is sensed via a $1\,\Omega$ resistor in series with the metal strips carrying the power. A spectrum analysis of both the voltage and the current showed components in these signals above $350\,\text{Hz}$. The anti-aliasing, second-order filter was designed with a cut-off frequency $f_c = 31\,\text{Hz}$. The Design-to-Cost approach, classic in high-volume manufacturing, led to the now unusual choice of a real-pole filter $G(s) = 1/(sRC+1)^2$, where $s$ is the Laplace variable, approximated by a Cauer Resistor Capacitor (RC) ladder network (Balabanian, 1958). The values R and C are chosen thanks to $f_c = 1/(2\pi RC)$. Moreover, the scaled values of the second RC network, $R/d$ and $Cd$ with $\{d \in \mathbb{R} : d > 0\}$, are computed to meet the specifications of the maximum magnitude error $e(d)$ between $G(s)$ and $G_a(s)$, the transfer function of the Cauer RC ladder defined by $G_a(s) = 1/\big((sRC)^2 + s(d+2)RC + 1\big)$. The value of $e(d)$ is given by equation (3). We chose $d = 0.1$ to have less than $0.5\,\text{dB}$ error, with no sensible impact on the later computations. An implementation with two identical RC sections (i.e. $d = 1$) would lead to $e(1) = 3.5\,\text{dB}$, which would degrade the overall performance. Both the voltage and the current are filtered by such ladder networks before being sampled at $f_s = 100\,\text{Hz}$:

as there are no components in the power spectrum between $f_s/2$ and 350 Hz, there is no aliasing.

$$e(d) = 20 \log_{10} \left( \frac{d+2}{2} \right) \tag{3}$$

The algorithms are written in C language and run in real-time on an Arduino Mega 2560 which has 8192 bytes of Random Access Memory (RAM). The analog signals are sampled and quantized by the integrated analog to digital converter in the microcontroller, with the sampling period defined by $t_s = 1/f_s$, and the sampling time being $kt_s$ with $k \in \mathbb{N}$.

The bijective case for the slot car relies on an three-step imitation procedure:

1. A human player first drives the car for $n$ laps, with $n = 3$ in our experiments.

2. The $K$ sampled voltages $v(kt_s)$ and currents $i(kt_s)$ of the shortest lap (with corresponding $t_{best}$ lap time) are stored in RAM for $0 \leq k < K$, to be replayed by the AI.

3. An optimization method (Newton) minimizes the difference between the AI's lap time and $t_{best}$ by scaling the recorded samples $v(kt_s)$ used to generate the Pulse-Width Modulation (PWM) control signal.

The analogy-based approach relies on two modules: the reward module, and the decision module described below.

As in traditional RL, our approach relies on a reward from the environment. This reward is based on three variables: the lap time (measured directly with the lap counter), the presence of the car on the track (binary information), and the fact that the car is moving (also binary information). The algorithm that we designed to provide this reward constantly monitors the car so as to detect that it did not crash (i.e. that it did not leave the track when the velocity was too high) or that it did not stop (when the current was too low to move the car). Both detectors are based on k-nearest neighbors algorithms (k-NN) applied to the voltage and the current. They are implemented as boolean tests on the signals after comparison with some thresholds, to speed up the execution of the algorithm on the microcontroller. As an illustration, a crash can be detected when the voltage is high and the current is near zero: it means that there is no more load (no DC motor) in contact with the strips, even though the voltage is still applied.

Using this reward model, the AI can successfully pilot the car on previously unencountered tracks. It does not replay scaled samples of any human driving. The only information reused by the algorithm is the safe speed: it does not trigger the "car crash" reward signal, yet it maintains the car in motion, thus not triggering the "car stop" reward signal. As the circuit is unknown, the bijective case cannot be used: there is no bijection between circuits. The algorithm only relies on the analogy-based approach and transposes knowledge previously acquired for a different circuit configuration thanks to equation (1). This knowledge – a safe speed for a given $s$ – is transposed via non-bijective analogies presented in 3.3 with the function $h((u,i)_s)$ evaluated with a classifier. Any classifier can be used, including unsupervised learning methods, as the two classes are clustered and separated. For simplicity, we used a k-NN.

In practice, the analogy-based approach starts on the unknown circuit with the safe speed. The algorithm infers in real time, from only current and voltage measurements, whether the car is in a configuration that we humans call either curve or straight. The algorithm then chooses the best control signal based on its previous experiences (best in order to reach the goal of decreasing lap time while staying on the track). Even though we use the terms "straight" and "curve" in our explanation, the algorithm simply classifies current and voltage to choose a control signal so as to stay on the track while decreasing the lap time. The algorithm uses this past knowledge (the control signal for each class) in a previously unencountered situation. In this way, it generalizes its strategy and adapts to a radically different case: circuit 2 differs from circuit 1, and a replay of a recorded strategy learned on one circuit or scaled recorded samples of the human driving would fail on the second circuit.

### 4.1.2 RESULTS

The experiments described in this article are conducted on two circuit configurations of different complexity presented in figure 1. The results of our experiments for the bijective and the analogy cases are summarized in table 1. Values are tabulated as the mean and the standard deviation from

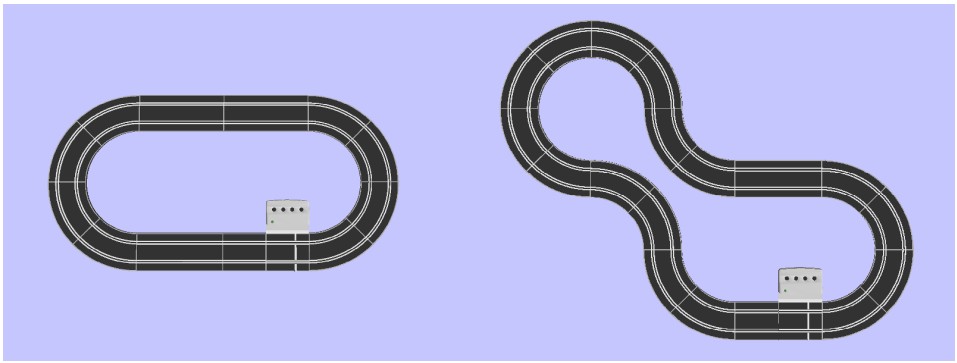

Figure 1: Slot car configurations: circuit 1 (left), circuit 2 (right).

Table 1: Lap times in seconds for two circuit configurations

|  | **HUMAN** | **AI** | **AI SETTINGS** |
|---|---|---|---|
| **CIRCUIT 1 (12 tracks)** | | | |
| First lap | $2.99 \pm 0.46$ | $3.12 \pm 0.09$ | PWM=39% of full speed |
| Final lap | $2.29 \pm 0.14$ | $2.52 \pm 0.08$ | Analogies (adaptive speed) |
| Best lap | $2.11$ | $2.12 \pm 0.05$ | Bijection (imitation) |
| | | | |
| **CIRCUIT 2 (18 tracks)** | | | |
| First lap | $4.30 \pm 1.16$ | $3.66 \pm 0.03$ | PWM=39% of full speed |
| Final lap | $3.08 \pm 0.54$ | $3.13 \pm 0.02$ | Analogies (adaptive speed) |
| Best lap | $2.67$ | $2.65 \pm 0.02$ | Bijection (imitation) |

the mean, except for the best lap which is the shortest lap time among all laps. We noticed that the first of eight consecutive laps is always the slowest one for the eight human subjects. The AI, which starts with no previous information, only relies on a safe speed as described in 4.1.1 using a constant PWM of 39% of the full speed. The analogy-based AI, which does not replay any recorded samples of a human driving, improves lap times in less than ten laps, even on an unknown track. On the longest and most complex circuit configuration (circuit 2), it almost ranks best, as tabulated on the line "Final lap". While the final human lap time is lower than the final AI lap time ($2.29\,\mathrm{s}$ vs $2.52\,\mathrm{s}$ for circuit 1, $3.08\,\mathrm{s}$ vs $3.13\,\mathrm{s}$ for circuit 2), the human unsurprisingly exhibits a higher standard deviation from the mean ($140\,\mathrm{ms}$ vs $80\,\mathrm{ms}$ for circuit 1, $540\,\mathrm{ms}$ vs $20\,\mathrm{ms}$ for circuit 2). Future improvements of the AI on the unknown track will include an optimization of the two speeds transposed by the function $h((u,i)_s)$: only a safe speed was used during our experiments, leading to no car crash for the AI, contrary to some laps by the humans and thus not taken into account.

Lastly, the bijective strategy – imitating the best human lap – also leads to the best lap time. However, contrary to the solution with analogies, it only works for an identical circuit. This means that while the best bijective (imitation) lap time ($2.65\,\mathrm{s}$) for circuit 2 is lower, thus better than the final lap time for the analogy (adaptive speed) AI ($3.13\,\mathrm{s}$), this strategy can only be used on circuit 2 and cannot lead to a generalization. It is only mentioned here as it gives an empirical lower bound for the lap time on a given circuit.

To summarize the slot car case, we implemented the theoretical method exposed in section 3.3 that allows the AI to reuse previously acquired knowledge on a new circuit where a replay of recorded samples of a human driving would lead to an immediate car crash. Even though there is no bijection between the different circuits, in practice this theory allows to generalize knowledge to any different circuit (within the limits imposed by the size of the available RAM).

## 4.2 ATARI 2600 GAMES

### 4.2.1 SETUP AND IMPLEMENTATION OF THE THEORY

While the slot car allowed us to validate the approach on real analog signals in a simple configuration, the ALE allowed us to validate the approach on more complex configurations while dealing with signals already sampled coming from the emulator. The concepts of entities with "Me" and life impulse introduced in 3.2 are also used to play Atari 2600 games. Our proof-of-concept is based on the detection of such entities thanks to image processing: Sobel operator (center image on figure 2) and bounding-box detection (right image on figure 2). It relies on the OpenCV (2017) library. The entity "Me" is found using system identification. Signals such as impulses and pseudorandom sequences (Levine, 2011) are sent to ALE to first detect the entities affected by these signals, then to build a dynamic model of the "Me". One or a few entities are controllable: they are the "Me". Their shapes can change during the gameplay, such as the paddle in Breakout, thus the possibility to identify different entities as the "Me". These measurements also update the probability functions $p(E, F)$ for entities $E$ and $F$ that the contact between these entities changes the score, in a way similar to the reward function in RL. From these functions $p$, friends and enemies are inferred, leading to a basic survival strategy outlined in 3.2.

The tests are carried out with the settings from Mnih et al. (2015): the AI plays for a maximum of 5 minutes. We choose to use the DQN as the reference: the reason being that this publication is one of the most cited in relation to Atari 2600 games, and is the de-facto benchmark to which one must refer. Although we aim to control cyberphysical systems, we needed to validate the versatility of our approach by first testing it on this standard. We fully replicated the setup using code made publicly available by the authors, and we obtained the same results as the publication. We were thus able to extract the score for the DQN for a low number of training frames, so as to compare with our approach.

### 4.2.2 RESULTS

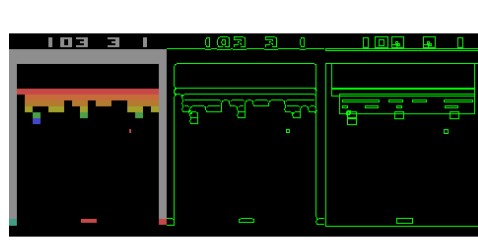

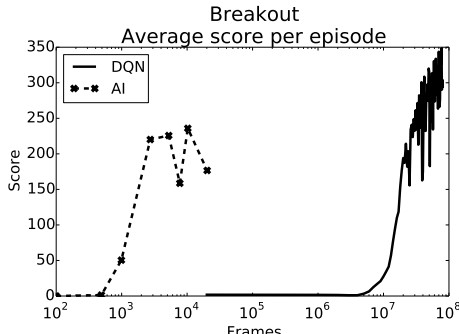

Figure 2: Image processing on Breakout          Figure 3: Scores for Breakout

Results[7] are presented in table 2 for a training time of 10 000 frames (less than 3 minutes), which is 20 000 times less frames than the average training standard reported in Mnih et al. (2015). While the DQN achieves better results with millions of training frames, our AI reaches decent scores with comparatively much fewer frames, as plotted for Breakout on figure 3.

A preliminary analysis of what really occurs while our algorithm learns is as follows: during the first thousand frames, the "me" is not yet correctly identified, as the digits, the ball and the paddle move randomly when controlled by the pseudorandom sequence. Once the identification has converged to the only system that the AI directly controls – the paddle, neither the ball nor the digits –, the algorithm looks for friends and enemies. It also detects that the ball is a friend, as it sometimes increases the score (when it breaks a brick).

---

[7]Results for Human and Random policies are reprinted from Mnih et al. (2015). The training period for the human player lasts 432 000 frames (2 hours).

Table 2: Comparison of game scores after $10\,000$ training frames

| GAME | RANDOM | HUMAN | DQN | AI |
|------|--------|-------|-----|-----|
| Breakout | 1.7 | 31.8 | 1.25±1.02 | 235.88±74.41 |
| Pong | −20.7 | 9.3 | −21.00±0.00 | −9.13±4.99 |

The best scores are in the range of 200 points which, after analysis, corresponds to partially destroyed rows of bricks. It never destroys all the bricks, as it sometimes misses the ball, especially when it looses the "me", for instance when the paddle's size changes or when the paddle disappears according to the basic image processing algorithm. Moreover, we noticed that the movement of the "me" under control of the algorithm sometimes never reaches a steady-state: it oscillates by a few pixels at a frequency of $5.4\,\mathrm{Hz}$. Our explanation from a control perspective is as follows: the "me" can be approximated by a second-order system, and the control strategy is almost equivalent to a proportional controller. This, in the context of Linear-Time-Invariant (LTI) systems, would already explain the oscillations. Moreover, the strong non-linearities present both in the control input and the non-LTI "me" also explain in part these oscillations. The input being quantized to only three values (left, nothing, right), the closed-loop system generates a signal similar to limit-cycles. These oscillations, in turn, are responsible for many of the balls missed by the algorithm.

To summarize the results on the Atari games, we implemented a few of the concepts presented in 3.1: the notion of entities rather than samples or pixels, and the "me" with the behavior introduced in 3.2. This led to a learning time of a few thousand frames to get a better than human score on Breakout, however it still does not match the best score reached by the DQN after millions of frames on Pong.

## 5  CONCLUSION AND FUTURE WORK

Continental philosophy lead us to formalize a mathematical concept to control an agent evolving in a world, whether it is simulated or real. The power of this framework was illustrated by the theoretical example of the slot car on unknown circuits. Results from experiments with a real slot car, using real analog signals confirmed our expectations, even though it only used a basic survival approach. Moreover, the same basic survival strategy was applied to two Atari 2600 games and showed the same trend: even though not as skilled as, for instance, DQN-based agents trained with two hundred million frames, our AI reached in less than ten thousand frames scores that DQN met after learning with a few million frames.

The next steps are to apply the transposition properties to the Atari games, as we did for the slot car, which should further decrease the learning time when playing a new game. Moreover, going beyond the basic survival strategy will be mandatory to reach higher scores: approaches based on Monte-Carlo Tree Search will be investigated.

ACKNOWLEDGMENTS

The authors wish to thank X. Xxxxxxxxx and X. Xxxxxxxxx for building the first slot car prototypes, X. Xxxxxxxxx for sharing his anthropologist views, and the anonymous reviewers for their feedback.

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
