# OpenReview forum: "Learning to play slot cars and Atari 2600 games in just minutes"
_ICLR.cc/2018/Conference — Reject_

### Official Review · AnonReviewer3 · 2017-11-26
**learning for Atari**

**Rating:** 3
**Confidence:** 2

**Review:**

The authors argue that many machine learning systems need a large amount of data and long training times.  To mend those shortcomings their proposed algorithm takes the novel approach of combining mathematical category theory and continental philosophy.  Instead of computation units, the concept of entities and a 'me' is introduced to solve reinforcement learning tasks on a cyber-physical system as well as the Atari environment. This allows for an AI that is understandable for humans at every step of the computation in comparison to the 'black box learning of an neural network.


Positives:
	•	Novel approach towards more explainable and shorter training times/ less data
	•	Solid mathematical description in part 3.3
	•	Setup well explained


Negatives:
	•	Use of colloquial language (the first sentence of the abstract alone contains the word 'very' twice)
	•	Some paragraphs are strangely structured
	•	Incoherent abstract
	•	Only brief and shallow motivation given (No evidence to support the claim)
	•.      Brief and therefore confusing mention of methods
	•	No mention of results
	•	Very short in general
	•	Many grammatical errors (wrong tense use, misuse of a/an,... )
	•	Related Work is either Background or an explanation of the two test systems. While related approaches in those systems are also provided, the section is mainly used to introduce the test beds
	•	No direct comparison between algorithm and existing methods is given. It is stated that some extra measures from other measures such as sensors are not used and that it learns to rank with a human in under a minute. However, many questions remain unanswered: But how good is this? How long do other systems need? Is this a valid point to raise? What score functions do other papers use?
	•	2.2: Title choice could have been  more descriptive of the subsection. 'Video Games' indicates a broader analysis of RL in any game but the section mainly restricts itself to the Atari Environment
	•	While many methods are mentioned they are not set in context but only enumerated. Many concepts are only named without explanation or how they fit into the picture the authors are trying to paint.

	•	A clear statement of the hypothesis and reason/motivation behind pursuing this approach is missing. Information is indirectly given in the third section where the point is raised that the approach was chosen in contrast to 'black box NNs'. This seems to be a very crucial point that could have been highlighted more. The achieved result are by no means comparable to the NN approaches but they are faster and explainable for a human.
	•	Dreyfus' criticism of AI is presented as the key initiator for this idea. Ideas by other authors that utilise this criticism as their foundation are conceptually similar, they could have therefore been mentioned in the related work section.
	•	The paper fails to mention the current movement in the AI community to make AI more explainable. One of their two key advantages seems to be that they develop a more intuitive explainable system. However, this movement is completely ignored and not given a single mention. The paper, therefore,  does not set their approach in context and is not able to acknowledge related work in this area.
	•	The section about continental based philosophy is rather confusing
	•	Instead of explaining the philosophy, analytical philosophy is described in details and continental philosophy is only described as not following analytical patterns. A clear introduction to this topic is missing.
	•	When described, it is stated that it's a mix of different German and French doctrines that are name dropped but not explained\ and leave the reader confused.
	•	Result section not well structured and results lack credibility:
	•	Long sections in the result section describe the actual algorithm. This should have been discussed before the results.
	•	Results for slot car are not convincing:
	•	Table 1 only shows the first the last and the best lap (and in most of them the human is better)
	•	Not even an average measure is given only samples. This is very suspicious.
	•	Why the comparison with DQN and only DQN? How was this comparison initialised? Which parameters were used? Neither is the term DQN resolved as Deep Q-Network nor is any explanation given. There are many methods/method classes performing RL on the Atari Environment. The mention of only one comparison  leaves reasonable doubt about the claim that the system learns faster.

SUMMARY: Reject. Even though the idea presented is a novel contribution and has potential the paper itself is highly unstructured and confusing and lacks a proper grammar check. No clear hypothesis is formed until section 3. The concept of Explainable AI which could have been a good motivation does not find any mentioning. Key concepts such as continental philosophy are not explained in a coherent way. The results are presented in a questionable way. As the idea is promising it is recommended to the authors to restructure the paper and conduct more experiments to be able to get accepted.

---

> ### Author Response · Authors · 2017-12-07
> **Results are given as follows: Mean +- (Standard deviation from the mean)**
>
> Dear AnonReviewer3,
> Thank you for the feedback. We are glad that you understood the target at which we were aiming: the reduction of training time when controlling devices. Below we will try to clarify the points that you mentioned.
>
> - “No direct comparison between algorithm and existing methods is given”. For the slot car setup, we only compared our results to published data as mentioned in 2.1 (example: twelve hours to learn to drive a slot car). We did not reproduce the complex hardware setup described in the publications, as our target, described in the introduction, was to find an “alternative approach to teach computers to learn quickly to perform as efficiently as the existing solution with approximately one percent of the training data, time, and computing resources”. Neither the vision-based approach nor the added-sensors and embedded processor solution fit in this framework. The results we obtained with our own algorithm are tabulated in 4.1 and need, as written in our text, up to ten laps to learn. This leads to less than a minute of learning time, as the longest laps are less than 6 seconds (table 1). Even though no lower-bound was given for the lap time (which would indeed give an indication of how good drivers are), the minute versus hours of learning time was in line with our initial claim. “How long do other systems need? Is this a valid point to raise?” is thus answered by figures in our paper, and is going towards our original goal that we described in the introduction “to perform as efficiently as the existing solution with approximately one percent of the training data, time, and computing resources”.
> - We agree that 2.2 could have mislead the reader into thinking that we would go beyond the Atari 2600 games (only a Nintendo game is mentioned in Lee et al., 2014).
> - “A clear statement of the hypothesis and reason/motivation behind pursuing this approach is missing”. The motivation is given in the introduction, along with the targets: “algorithms that control cyberphysical systems to learn with very little data how to operate quickly in a partially-known environment” and the target is “teach computers to learn quickly to perform as efficiently as the existing solution with approximately one percent of the training data, time, and computing resources”.
> - About bibliographical references and Dreyfus: we are willing to add more references, including both explainable AI and other approaches relying on Dreyfus ideas. We only removed them from our submission to comply with the “strong recommendation” of 8+1 pages.
> - Continental based philosophy definition: we followed the classic explanation of continental philosophy as defined with respect to analytical philosophy. It is difficult to provide a definition of continental philosophy that is accepted by everyone in the field, except by opposition to analytical philosophy.
> - “Result section not well structured and results lack credibility”: if allowed to go beyond the 8+1 page, we will be glad to split the description of the experimental setup and the results. As proposed to other reviewers, we will then also spend more time explaining the results that are tabulated, and describing what really occurs while our algorithm is learning. Regarding Table 1, we mostly gave average times plus or minus the standard deviation. Example for circuit 2: 3.08+-0.54s for the human means 3.08s average, and a standard deviation from the mean (std) of 0.54s. The AI reached 3.13s for the mean value, which is worse than 3.08s for the human, however with a std of 0.02s compared to 0.54s for the human. The AI is thus more consistent than a human, while being slightly slower than the human. “Not even an average measure is given only samples. This is very suspicious”: the only samples are the best lap for the human being. All other results consist of an average and a standard deviation.
> - “Why the comparison with DQN and only DQN? How was this comparison initialised? Which parameters were used?” We agree on the lack of explanation for the term DQN, this was an oversight on our part. The comparison with the DQN was done following the publication referenced in our paper, reproducing the results using the same setup as explained in 4.2: “The tests are carried out with the settings from Mnih et al. (2015)”. As for why we choose the DQN, the reason is that this publication is one of the most cited in relation to Atari 2600 games, and is the de-facto benchmark to which one must refer. Although we aim to control cyberphysical systems, we needed to validate the versatility of our approach by first testing it on this standard.
>
> As asked to the other reviewers, please tell us if adding a few pages to clarify the setup and the results could lead to an accepted paper. It would also include minor modifications to remove any ambiguity found by yourself and the other reviewers, including grammatical errors.
>
> Sincerely yours,
> The Authors.

---

### Official Review · AnonReviewer1 · 2017-11-28
**A new approach to AI based on concepts from continental philosophies**

**Rating:** 2
**Confidence:** 5

**Review:**

In this paper the authors address the very important challenges of current deep learning approaches, which is that these algorithms typically need an extraordinarily large number of training rounds to learn their strategies.  The authors note that in real life, this type of training will outstrip both the training and time budget of most real world problems.  The solution they propose is to take a high level approach and to learn more like humans do by creating strategies that involve relationships between entities rather than trying to build up strategies from pixels.
The authors credit their reframing of their approach to AI to the “continental philosophers” (e.g. Heidegger) in opposition to the “analytical philosophers” such as Wittgenstein.  The authors associate current machine learning approaches with the analytic philosophers, based on propositions that are either provably true or untrue and their own approach as in opposition to these, however from my reading of this paper what the authors are saying is that if you start learning with higher level concepts (relationships between entities) rather than doing analysis on low level information such as pixels.   Starting with low level concepts makes learning very difficult at first and leads to a path where many trials are required.  Staring from higher level concepts such as relationships between entities allows learning to happen quickly and in a manner much more similar in nature to what humans actually do.
While the authors bring up many valid points, and in essence I believe that they may be correct, the flaw in this paper is that they do not provide methods for teaching computers to learn these higher level concepts.  The algorithms they present all require human knowledge to be encoded in the algorithms to identify the higher level concepts.  The true power of the deep learning approach is that it can actually learn from low level data, without humans hand crafting the higher level entities on their behalf.

While I agree with Dreyfus that understanding what is important and interesting given a situation would be an incredible boon to any AI algorithm, it remains an unsolved problem as to how to teach a computer to understand what is interesting in a scene with the same intuition that a human has.  In the first experiment the authors need to pre-define the concepts of a straight road and a curved road and identify them for the algorithm.  They also need to tell the algorithm exactly how to count the number of sections that the track has.  In the second experiment, to identify the “Me” in the game, the authors instruct the computer to recognize “me” as the things that move when the controller is activated.  While in some ways this is clever, mimicking what a child might do to see what moves in the world when it issues a command to move from its own brain and thus learning what “me” is, children take year to develop a sense of “self” and part of that is learning that a “concept of self” is an interesting and useful thing to have.  In their work the authors know, from their human intelligence, what are the important concepts in the game (again from a human perspective) and devise simple methods for the computer to learn these.  Again the problem here is that the human has to define the important concepts for the computer and define a specific strategy for the computer to learn to identify these important policies.  Data intensive deep learning algorithms are able to infer strategies without these concepts being defined for them.

This reframing does point out a different and perhaps better path for AI, but it is not entirely new and this paper does not present a method for getting from sensed data to higher level concepts.  For each of the experiments, the strategies used rely on human intuition to define policies.  In the first experiment with slot cars, a human needs to provide n laps of driving to imitate.  The authors identify the “shortest lap” and store it for the “AI” to replay.  The only “learning” is from an optimization that minimizes the difference between the AI’s lap time and the best lap time (tbest) of the human by scaling that recorded sample of the human driving.  This results is a strategy that is essentially just trying to replicate (imitate) what the human is doing will not lead to a generalizable learning strategy that could ever exceed a human example.   This is at best a very limited form of imitation learning.  The learning process for the second example is explained in even less detail.
Overall, this paper presents a different way of thinking about AI, one in which the amount of training time and training data required for learning is greatly reduced, however what is missing Is a generalizable algorithmic strategy for implementing this framework.m

---

> ### Author Response · Authors · 2017-12-07
> **Section 3.3 describes the theory used when there is no "imitation learning" and illustrates it with the slotcars**
>
> Dear AnonReviewer1,
> Thank you for the feedback. We are glad that you understood the target at which we were aiming: the reduction of the training time when controlling devices, and the concept we borrowed from different sciences such as philosophy and linguistics. We hope to clarify the fact that the algorithm does not need a specific training for each configuration, and that it does not always replay or “scales recorded samples of the human driving” as you mentioned.
>
> We illustrate two cases with the slot car: the case where the car drives on the same track (bijective case), and the case where the track is unknown (analogy case). While it is true that the algorithm learns from the best lap for the bijective case, as you clearly describe in your review, the analogy case is different. In the analogy case, as written on page 7, section 4.1, the algorithm “transposes knowledge previously acquired for a different track configuration”. As there is no bijection between the two circuit configurations, there is no possibility to replay something that would have been recorded. The algorithm infers in real time, from only current and voltage measurements, that the car is in a configuration that we (humans) call curve or straight. It relies on a classifier (k-nn) with two classes. This number of classes could be expanded to higher values, that would lead to a more complex description in human terms, which would in turn defeat the purpose of this toy problem. The algorithm then chooses the best control signal based on its previous experiences (best in order to reach the goal of decreasing lap time while staying on the track). We do admit that we used the terms “straight” and “curve” in our explanation, but the algorithm simply classifies current and voltage to choose a control signal so as to stay on the track while decreasing the lap time.
>
> The algorithm uses this past knowledge (the control signal for each class) in a previously unencountered situation. In this way it generalizes its strategy and adapts to a radically different case: circuit 2 differs from circuit 1, and a replay of a recorded strategy learned on one circuit or scaled “recorded samples of the human driving” would fail on the other circuit.
>
> The only shortcoming we did when applying this theory for the slotcar was to skip the search for the “me”, as there is only one entity with dynamic behavior. The algorithm still applies concepts outlined in 3.2 such as the search for enemies (the enemy being a car crash).
>
> However, our algorithm does not “count the number of sections that the track has”: please tell us what part of our document could be improved to avoid such misunderstanding. The algorithm has absolutely no way to count such sections, nor the required sensors as far as we can tell: it only measures the voltage and the current.
>
> The second example (Atari games) does not even rely on the bijective case, because there is no human-provided reference or gameplay. It is thus discovering everything, as in classic reinforcement learning approaches. The only hard-coded concepts are:
> - the fact that there is at least one “me” among the entities (which means, in control theory terms, that there is at least one system responding to a control signal. Its transfer function is unknown).
> - the fact that going towards friends is the first strategy to apply to survive.
> The rest is inferred by the algorithm: the friends and the enemies are updated based on the evolution of the score function, and the control signal sent is based on this information. This is the reason why it needs a few thousands frames to start increasing the score: in the beginning, it does not properly locate the “me” or, if it does, it does not yet know who is a friend. Once it is inferred (as it is inferred with deep learning or reinforcement learning methods), only the concepts relevant for the task at hand are transferred, even if there is no bijection between the structures and the states (as explained in section 3.3).
>
>
> We are willing to add more references, including both explainable AI and other approaches relying on Dreyfus’ ideas. We only removed them from our submission to comply with the “strong recommendation” of 8+1 pages. We can also shorten the description of the analog electronics (which we included because of the added constraint of low cost when designing this kind of new solution). We could thus spend more time explaining the results that are tabulated, and what really occurs when our algorithm is learning.
>
> Please tell us if such modifications are within the scope of what is advised (including adding a few pages for the explanation of the results), and if it could lead to an acceptation of the paper.
>
> Sincerely yours,
> The Authors.

---

### Official Review · AnonReviewer2 · 2017-12-01
**Unusual, perhaps creative paper, very cryptically presented-- hard to evaluate fairly for this reviewer**

**Rating:** 3
**Confidence:** 1

**Review:**

For me, this paper is such a combination of unusual (in the combinations of ideas that it presents) and cryptic (in its presentation) that I have found it exceedingly hard to evaluate fairly. For example, Section 4 is very unclear to me. Its relationship to Section 3.3 is also very unclear to me.

Before that point in the paper, there are many concepts and ideas alluded to, some described with less clarity than others, but the overall focus is unclear to me and the relationship to the actual algorithms and implementation is also unclear to me. That relationship (conceptual motivation --> implementation) is exactly what would be needed in order to fully justify the inclusion (in an ICLR paper) of so much wide-ranging philosophical/conceptual discussion in the paper.

My educated guess is that other reviewers at ICLR may have related concerns. In its current state, it therefore does not feel like an appropriate paper for this particular conference. If the authors do feel that the content itself is indeed a good fit, then my strong recommendation would be to begin by re-writing it so that it makes complete sense to someone with a "standard" machine learning background. The current presentation just makes it very hard to assess, at least for this reviewer.

If other reviewers are more easily able to read and understand this paper, then I will be glad to defer to their assessments of it and will happily retract my own.

---

> ### Author Response · Authors · 2017-12-06
> **Motivation, theory and results**
>
> Dear AnonReviewer2,
> Thank you for your feedback. We hope to clarify the points that you mentioned in your review:
> - the motivation is outlined in the introduction (page 1) with the target being “algorithms that control cyberphysical systems”, under the constraint “to learn with very little data how to operate quickly in a partially-known environment”. The quantitative target was given by “This work thus started as an alternative approach to teach computers to learn quickly to perform as efficiently as the existing solution with approximately one percent of the training data, time, and computing resources.”
> - The concepts, borrowed from existing and known results in sciences such as structural linguistics and continental philosophy, are described in section 3 “continental-philosophy-based theoretical approach”. We acknowledge the fact that continental-philosophy is defined in our paper as (in a simplified form) “everything but analytic philosophy” for the lack of better definition that is accepted by everyone in the field.
> - Section 3.3 deals with the case when imitation does not work, i.e. when the AI cannot reach the goal by imitating a known, working solution. The results from this section are used in the experimental part 4, for instance for the slot car as written on page 7 in 4.1: “The analogy-based approach transposes knowledge previously acquired for a different track configuration thanks to equation (1). This knowledge – a high safe speed for a given s – is transposed via non-bijective analogies presented in 3.3 with the function h((u, i) s ) evaluated with a k-NN.”
>
> As proposed to the other reviewers, if the 9-page limit (that was strongly recommended) can be bypassed, we will gladly describe and analyze more deeply the results that we tabulated in our paper, including when imitation is used, and when analogies are used.
>
> Please tell us if such modifications would make our paper in line with ICLR requirements.
> Regards,
> The Authors

---

> > ### Comment · AnonReviewer2 · 2017-12-07
> > **thanks for the response**
> >
> > Thank you for the response.
> >
> > Also, I should add that as a reviewer, I do appreciate that the authors respected the page limit.
> >
> > In terms of whether it's worthwhile for the authors to invest time in modifying the paper by exceeding the limit in order to explain things more clearly, unfortunately  I don't know the answer to that.
> >
> > I suspect that for this particular conference, it may not be worth the time, but I acknowledge that I may be biased due to not fully understanding the paper in its current form. From what I see from the now-posted other reviews, it may simply make it clearer that the paper does not quite fit this conference.But until the details are provided, I just don't know.
> >
> > On the other hand, I believe that expanding the (machine learning) details of the paper will be helpful for publication in any machine-learning venue, so in that sense, if the authors are particularly interested in publishing this work in a machine-learning venue,it is indeed worth providing more such details.
> >
> > As one possible guideline, I would suggest that the authors imagine if they were to hand the paper to a competent graduate student, and ask: would the student have enough information, in what is provided in the paper, to implement something close to what the authors have done?

---

### Decision · Program_Chairs · 2018-01-29
**ICLR 2018 Conference Acceptance Decision**

**Decision:**

Reject

**Comment:**

This paper does not seem completely appropriate for ICLR.